# Urinary Uremic Toxin Signatures and the Metabolic Index of Gut Dysfunction (MIGD) in Autism Spectrum Disorder: A Stool-Phenotype-Stratified Analysis

**DOI:** 10.3390/ijms262110475

**Published:** 2025-10-28

**Authors:** Joško Osredkar, Teja Fabjan, Kristina Kumer, Maja Jekovec-Vrhovšek, Joanna Giebułtowicz, Barbara Bobrowska-Korczak, Gorazd Avguštin, Uroš Godnov

**Affiliations:** 1Institute of Clinical Chemistry and Biochemistry, University Medical Centre Ljubljana, Zaloška cesta 2, 1000 Ljubljana, Slovenia; josko.osredkar@kclj.si (J.O.); teja.fabjan@kclj.si (T.F.); kristina.kumer@kclj.si (K.K.); 2Faculty of Pharmacy, University of Ljubljana, Aškerčeva 7, 1000 Ljubljana, Slovenia; 3Center for Autism, Unit of Child Psychiatry, University Children’s Hospital, University Medical Centre Ljubljana, 1000 Ljubljana, Slovenia; maja.jekovec@kclj.si; 4Department of Bioanalysis and Drug Analysis, Faculty of Pharmacy with the Laboratory Medicine Division, Medical University of Warsaw, Banacha 1, 02-097 Warsaw, Poland; joanna.giebultowicz@wum.edu.pl; 5Department of Toxicology and Food Science, Faculty of Pharmacy with the Laboratory Medicine Division, Medical University of Warsaw, Banacha 1, 02-097 Warsaw, Poland; barbara.bobrowska@wum.edu.pl; 6Department of Microbiology, Biotechnical Faculty, University of Ljubljana, Groblje 3, 1230 Domžale, Slovenia; gorazd.avgustin@bf.uni-lj.si; 7Natural Science and Information Technologies, Faculty of Mathemathics, University of Primorska, Gljagoljaška ulica 8, 6000 Koper, Slovenia

**Keywords:** autism spectrum disorder, uremic toxins, microbiota–host interactions, p-cresyl sulfate, indoxyl sulfate, gut metabolic dysfunction, urinary biomarkers, Bristol Stool Chart

## Abstract

Gut-derived uremic toxins may play a key role in neurodevelopmental conditions such as autism spectrum disorder (ASD) via host-microbe metabolic interactions. We evaluated five uremic toxins—p-cresyl sulfate (PCS), indoxyl sulfate (IS), trimethylamine N-oxide (TMAO), asymmetric dimethylarginine (ADMA), and symmetric dimethylarginine (SDMA)—in urine samples of 97 children with ASD and 71 neurotypical controls, stratified by Bristol Stool Chart (BSC) consistency types. Four of these toxins (PCS, IS, TMAO, ADMA) were integrated into a novel composite biomarker called the Metabolic Index of Gut Dysfunction (MIGD), while SDMA was measured as a complementary renal function marker. While individual metabolite levels showed no statistically significant differences, group-wise analysis by stool phenotype revealed distinct trends. ASD children with hard stools (BSC 1–2) showed elevated PCS levels and the MIGD score (median 555.3), reflecting phenolic fermentation dominance with reduced indolic detoxification. In contrast, children with loose stools (BSC 6–7) had the lowest MIGD values (median 109.8), driven by higher IS and lower ADMA concentrations, suggestive of enhanced indole metabolism. These findings indicate that MIGD may serve as a novel biomarker to stratify metabolic phenotypes in ASD, linking urinary metabolite patterns to gut function. Further validation in larger and longitudinal cohorts is warranted to confirm its potential utility in precision microbiota-targeted interventions.

## 1. Introduction

The complex neurodevelopmental disorder known as autism spectrum disorder (ASD) is marked by a high frequency of gastrointestinal (GI) disorders, such as constipation, diarrhea, and changed stool consistency, in addition to behavioral and cognitive abnormalities [1]. It is becoming more widely acknowledged that these symptoms are essential to the illness rather than secondary comorbidities. Through immunological, endocrine, and metabolic pathways, the gut microbiota has been identified as a key facilitator of this brain–gut interaction capable of impacting neurodevelopment [2].

One important class of microbial-derived metabolites implicated in systemic and neurological health is the uremic toxins [3]. These compounds originate from bacterial metabolism of dietary precursors such as tryptophan, tyrosine, and choline [4,5]. Despite being first investigated in chronic renal illness, uremic toxins are now known to have a broad impact on immunological signaling, oxidative stress, endothelial integrity, and neuroinflammation. These processes are highly pertinent to the pathogenesis of ASD, and motivate further investigation of the role of uremic toxins in this disorder [6,7,8]. To our knowledge, this is the first study to evaluate PCS, IS, TMAO, and ADMA together as a panel of gut-derived and host-derived uremic toxins in ASD, and to integrate them into a composite biomarker. Previous reports have typically examined individual metabolites in isolation, without exploring their combined utility for stratifying ASD subtypes.

In this work, we investigate four uremic toxins—indoxyl sulfate (IS), p-cresyl sulfate (PCS), trimethylamine N-oxide (TMAO), and asymmetric dimethylarginine (ADMA)—as potential biomarkers for ASD pathogenesis. These four metabolites capture complementary aspects of microbial fermentation (phenolic, indolic, and amine pathways) and host detoxification pathways, enabling integrated analysis of gut–brain–metabolite interactions relevant to ASD pathophysiology.

PCS and IS are the microbial fermentation products of tyrosine and tryptophan metabolism, respectively [9,10]. PCS is produced by phenolic fermentation and has been demonstrated to enhance oxidative damage, activate the aryl hydrocarbon receptor (AhR), and weaken vascular and epithelial barriers ** [11]**. IS is an indolic molecule, and depending on its quantity and local context, can operate both as a mucosal immune regulator and a neurotoxin ** [12]**. TMAO is a hepatic oxidation product of microbially produced trimethylamine from choline or carnitine and has been associated with systemic inflammation, cardiometabolic risk, and possibly neurobehavioral traits [13,14]. Finally, ADMA is an endogenous nitric oxide synthase inhibitor that reflects methylation load and host cellular stress [15]. These substances enter the bloodstream and are eliminated by the kidneys, making urinary uremic toxin levels a non-invasive indicator of gut metabolic output [16].

The selection of PCS, IS, TMAO, and ADMA as uremic toxins of interest in ASD was guided by their known involvement in oxidative stress, endothelial dysfunction, and neuroimmune modulation [9,10]. Both PCS and IS have been implicated in autism-related research. Persico and Napolioni [17] highlighted p-cresol and its sulfated conjugate PCS as elevated in children with ASD, possibly linked to repetitive behaviors and detoxification imbalance [17]. Furthermore, PCS-glucuronide, an alternative conjugate excreted in urine alongside PCS-sulfate, has been identified as a marker of gut microbial activity and phenolic burden [18]. Similar findings apply to IS, which exists in both sulfated and glucuronidated forms, though IS-sulfate predominates in systemic circulation and is often used as a surrogate of microbial indole production [19]. ADMA serves as a readout of cellular stress and detoxification burden, which may be altered in children with ASD due to oxidative imbalance or impaired endothelial regulation [20].

The interplay between microbial fermentation and host detoxification is central to understanding systemic metabolite patterns and their potential impact on neurodevelopment [21]. Stool consistency (as determined by the Bristol Stool Chart) and water content are functional characteristics that reflect gut motility and microbial activity. Fermentation time and the formation of microbial metabolites are influenced by stool form, and hydration level. For example, faster transit tends to promote phenolic fermentation (e.g., PCS) while suppressing indolic metabolism (e.g., IS) [22]. Therefore, stool consistency is a biologically plausible stratification framework for understanding the effects of uremic toxins on ASD pathogenesis [23,24,25].

To illustrate the conceptual framework of this study, Figure 1 summarizes the interconnected microbial and host pathways leading to the formation of the key uremic toxins evaluated—p-cresyl sulfate (PCS), indoxyl sulfate (IS), trimethylamine N-oxide (TMAO), asymmetric dimethylarginine (ADMA), and symmetric dimethylarginine (SDMA). These metabolites arise from the microbial degradation of aromatic amino acids and host methylation processes, followed by hepatic conjugation and renal excretion. The schematic also highlights how variations in gut transit time and stool consistency can modulate these pathways, ultimately shaping the composite metabolic profile quantified by the Metabolic Index of Gut Dysfunction (MIGD).

Recent research supports the use of functional ratios such as IS/PCS and PCS/TMAO rather than absolute toxin levels as more informative indicators of microbial–metabolic balance. These ratios take into consideration changes in hepatic transformation, systemic clearance, and fermentation routes, all of which can differ among ASD subtypes [26,27,28]. The construction of MIGD was guided by the following biological rationale: First, the PCS/TMAO ratio captures the balance between tyrosine-derived phenolic fermentation and choline-derived methylamine metabolism, both of which are influenced by gut transit time and microbial composition. Elevated PCS production is associated with proteolytic bacterial activity in the distal colon, particularly under conditions of slow transit, while TMAO reflects dietary trimethylamine precursors and hepatic FMO3 oxidation capacity. Second, the IS/ADMA ratio reflects the interplay between tryptophan-derived indole production (a marker of bacterial tryptophanase activity) and endogenous asymmetric dimethylarginine as a marker of endothelial dysfunction. The composite ratio [(PCS/TMAO)/(IS/ADMA)] thus synthesizes phenolic dominance, indolic pathway efficiency, and host detoxification into a single index. Higher MIGD values indicate phenolic overload with suppressed indolic pathways, reflecting an imbalance toward slower gut transit and increased toxin retention. Conversely, lower MIGD values may reflect enhanced indole metabolism or more rapid intestinal transit, facilitating dilution and clearance of microbial metabolites [29,30,31,32,33].

In this work, we developed a novel composite biomarker, the Metabolic Index of Gut Dysfunction (MIGD), to capture meaningful information from this complexity. MIGD may improve our capacity to categorize ASD patients according to metabolic profile and predict the intensity of symptoms or responsiveness to treatment by reflecting phenolic overload, indole suppression, and uremic burden.

By establishing a connection between uremic toxin profiles and stool consistency and hydration, our work explores these new indicators of microbial metabolism in ASD. We evaluate conventional toxin concentrations, novel functional ratios, and the possibility of MIGD as a comprehensive marker of gut metabolic dysfunction in autism.

## 2. Results

We measured uremic toxin levels in urine (Table 1) and computed a composite index, the Metabolic Index of Gut Dysfunction (MIGD), to investigate metabolic abnormalities. MIGD reflects integrated metabolic burden (PCS/TMAO relative to IS/ADMA). This index is shown in Table 2 for both ASD and control children across stool-consistency-stratified subgroups, revealing unique metabolic profiles linked to phenotypic heterogeneity. In addition to PCS, IS, TMAO, and ADMA, we also quantified symmetric dimethylarginine (SDMA). SDMA is a structural isomer of ADMA, derived from protein arginine methylation, and is primarily excreted by the kidneys. Unlike ADMA, which inhibits nitric oxide synthase, SDMA reflects renal clearance and methylated arginine turnover [34]. Although SDMA was not incorporated into the composite index, its measurement provides complementary context regarding host detoxification and excretory function. We therefore report SDMA descriptively in the Results but restrict biomarker index calculations to PCS, IS, TMAO, and ADMA based on biological plausibility.

Urinary concentrations of the five uremic toxins—ADMA, SDMA, TMAO, IS, and PCS—stratified by study participant stool consistency are presented in Table 1. While no statistically significant differences were observed between controls and ASD subgroups (all *p* > 0.05), several directional trends emerged that support the hypothesis of gut–metabolite interactions influenced by transit time and microbial fermentation.

Among children with ASD, those with loose stools (BSC 6–7) exhibited the lowest median ADMA and highest IS concentrations, suggesting a possible shift toward indole-based tryptophan metabolism or reduced clearance due to accelerated intestinal transit. In contrast, the highest PCS levels were observed in ASD participants with hard stools (BSC 1–2), aligning with a phenolic fermentation dominance under slower gut motility conditions. TMAO values tended to be higher in controls and somewhat reduced across all ASD subgroups, particularly in those with altered stool form, consistent with reduced choline metabolism or altered hepatic oxidation.

Although small subgroup sizes—particularly in the BSC 6–7 group—limit statistical power and preclude definitive conclusions, the observed metabolic gradients across stool phenotypes are biologically plausible. These findings underscore the relevance of stool consistency as a functional parameter influencing microbial–host toxin production and excretion in ASD.

Table 2 shows median values (Q1, Q3) of the Metabolic Index of Gut Dysfunction (MIGD) for healthy controls and ASD participants stratified by Bristol Stool Chart (BSC) subgroups: hard stools (BSC 1–2), loose stools (BSC 6–7), and normal stools (BSC 3–5). *p* values reflect pairwise comparisons to the control group and total comparison across ASD subgroups (p TOT). Trends in metabolic imbalance were more apparent for MIGD than for FSI, though differences did not reach statistical significance.

Children with ASD and hard stools (BSC 1–2) exhibited the highest MIGD values, reflecting elevated PCS/TMAO ratios alongside relatively low IS/ADMA values. This combination suggests dominant phenolic fermentation and limited indolic detoxification, a metabolic profile previously associated with slower transit and increased gut-derived toxin retention. In contrast, those with loose stools (BSC 6–7) showed the lowest MIGD scores, driven by markedly elevated IS/ADMA ratios, which may indicate enhanced tryptophan turnover or altered excretory kinetics (Table 2). While overall group comparisons did not reach statistical significance (*p* = 0.088), these distinct directional shifts support the utility of MIGD as a sensitive integrative marker of gut metabolic dysfunction.

Ratios shown use group median concentrations from Table 1. For statistical significance, see individual-level analysis in Table 2. Calculated values of the Metabolic Index of Gut Dysfunction (MIGD) across study groups, based on median urinary toxin ratios: phenolic (PCS/TMAO) and indolic (IS/ADMA) pathways. MIGD captures the combined burden of dysregulated fermentation and host detoxification. Higher MIGD values indicate phenolic dominance with reduced indole detoxification.

The ratios that make up the Metabolic Index of Gut Dysfunction (MIGD) are given in Table 3. The ratio of methylamine-based microbial metabolism (TMAO from choline and carnitine) to phenolic fermentation (PCS from tyrosine metabolism) is shown in PCS/TMAO. Slower transit or increased proteolytic/phenolic activities are indicated by a high PCS/TMAO ratio. The equilibrium between indolic metabolites (IS from tryptophan breakdown) and host detox load (ADMA, associated with nitric oxide regulation and renal clearance) is captured by IS/ADMA. Increased tryptophan fermentation, compromised hepatic detoxification, or endothelial dysfunction could all be indicated by a high IS/ADMA ratio.

The ASD group with firm stools (BSC 1–2) had the greatest PCS/TMAO ratio, indicating strong phenolic fermentation (Table 3). Conversely, children with ASD who had loose stools (BSC 6–7) had the greatest IS/ADMA ratio, which is associated with increased or maintained indole production (Table 3). Because of the inhibitory effect of a high IS/ADMA denominator, MIGD was lowest in this group. The ASD hard stool subgroup had the highest MIGD, which was indicative of low indole detox efficiency and double elevation in phenolic dominance (Table 3). These findings demonstrate the power of MIGD and its constituent parts to offer distinct and complementary insights into the metabolic profiles of microbes and hosts in ASD. While individual metabolite levels showed no statistically significant differences (all *p* > 0.05), consistent directional trends emerged through composite index analysis. These preliminary findings warrant validation in larger, longitudinal cohorts with parallel microbiome assessment.

## 3. Discussion

Our research provides preliminary evidence for a metabolic relationship between stool consistency and gut-derived uremic toxins in children with autism spectrum disorder (ASD), characterized by consistent directional trends despite the absence of statistical significance. We linked gut fermentation profiles to gastrointestinal function through a novel composite biomarker based on urinary levels of PCS, TMAO, IS, and ADMA, called the Metabolic Index of Gut Dysfunction (MIGD).

Unexpectedly, our results indicated that children with ASD who had hard stools (BSC 1–2) had the highest MIGD, whereas children with loose stools (BSC 6–7) had the lowest value. MIGD is calculated using the PCS/TMAO ratio, which indicates phenolic dominance, and the IS/ADMA ratio, which indicates indole metabolism and detoxifying load. Our findings, therefore, involved the interaction of component ratios rather than just absolute amount of uremic toxins. We propose that functional ratios may capture biologically meaningful differences even when absolute metabolite levels do not reach statistical significance, thereby offering a more sensitive approach to subgroup characterization.

Children with ASD who had firm stools and high MIGD had low IS/ADMA and high PCS/TMAO ratios. These individuals’ gut metabolic phenotype is likely characterized by an excess of phenolic load and ineffective indolic pathways, which are linked to oxidative stress and weakened mucosal immunity [28,35]. On the other hand, children with loose stools and low MIGD had high IS/ADMA ratios, which may indicate increased indole synthesis or hindered elimination in the context of rapid gut transit [36]. Our findings extend those of Osredkar et al. (2023) [8], adding Bristol stool phenotype as a modifier of uremic toxin profiles. Constipated children showed the highest MIGD values (median 555.3), suggesting amplified phenolic fermentation and toxin load, consistent with literature on slow gut transit’s impact on colonic fermentation. Our reduced urinary TMAO in ASD, particularly in constipation, may initially contrast with literature on elevated plasma TMAO in ASD. This likely reflects compartmentalization: renal, hepatic, and microbial factors differ. Reduced TMAO here may result from decreased microbial TMA formation, restrictive diet, hepatic oxidation changes, or altered renal handling. Multi-compartment measurements in future studies will clarify this relationship [37].

Despite being a mathematically composite metric, MIGD has the potential to magnify non-linear interactions between host processing and microbial pathways. Therefore, MIGD should be understood in conjunction with its component ratios, especially when liver, kidney, or microbiome function may change. MIGD is meant to be a functional indicator of host responsiveness and gut microbial metabolic output rather than a stand-in for symptom intensity. The absence of statistically significant differences warrants careful interpretation. First, the small size in the BSC 6–7 group limits power. A post hoc analysis suggests 64/group is needed for moderate effects. Second, biological variability in toxin concentrations is high, due to variances in diet, hydration, renal clearance, and microbiota. Third, composite indices like MIGD may be more sensitive to metabolic differences. Therefore, patterns observed should be considered hypothesis-generating, with future validation needed in larger cohorts [38,39].

These results are also consistent with other observations that the microbiota of ASD is dominated by proteolytic and phenolic fermentation [40,41,42,43]. Our findings, however, provide granularity by demonstrating how stool consistency affects toxin expression and by providing reliable composite measures that may be used in intervention trials or longitudinal surveillance. Future studies are needed to explore the applications of this index, look into connections with behavior and diet, and evaluate how useful it is for classifying ASD subtypes or developing microbially focused treatments. We further suggest that MIGD be referred to by the alternate name “Osredkar-Godnov Index” (OGI).

### 3.1. Extended Applications of MIGD/OGI

The MIGD/OGI has a variety of potential uses in translational ASD research and therapeutic integration beyond metabolic classification. This indicator might be useful in early detection models, especially in baby cohorts at risk, when behavioral signs are preceded by microbial metabolic imbalance. Because of its dynamic character, it can be used to track the effectiveness of pharmaceutical treatments or interventions controlled by the microbiota. MIGD/OGI could aid in defining fermentation patterns particular to different species when combined with microbiome sequencing. MIGD/OGI also has potential as a transdiagnostic tool, which could help differentiate ASD from other gastrointestinal or neurodevelopmental disorders. The biomarker’s’ significance for precision diagnostics and tailored care pathways could be increased by further integrating it with neuroimaging and exposome data inside systems biology frameworks.

### 3.2. Strengths

The use of a large and clinically well-characterized cohort of children with autism spectrum disorder (ASD) and matched neurotypical controls is a strength of our research. The Bristol Stool Chart’s stool consistency stratification adds a functional gastrointestinal feature that makes it possible to identify metabolic trends unique to phenotypes. We also develop a unique method for incorporating numerous uremic toxins into a single metric for biologically meaningful interpretations (MIGD/OGI). With the potential for therapeutic use and longitudinal tracking, this urine biomarker offers a quantitative, non-invasive evaluation of host detoxification burden and systemic microbial fermentation activity.

Although SDMA was not included in MIGD, its role as a renal marker was assessed. SDMA is not a direct mediator of neuroimmune dysfunction, distinguishing it from ADMA. No significant differences in SDMA suggests minimal renal confounding in our other analytes [44,45].

SDMA values did not differ significantly between ASD and controls or across stool phenotypes, suggesting that renal clearance capacity was not a major confounder of the observed differences in other uremic toxins. Future work might examine SDMA in parallel with renal function markers to clarify whether subtle changes in arginine methylation or excretory load contribute to metabolic heterogeneity in ASD.

### 3.3. Limitations

This study has limitations: (1) Cross-sectional design precludes causation. (2) Small subgroups, notably for loose stool phenotype, limit power. (3) No dietary assessment; eating patterns may influence toxins. (4) No microbiome data limits mechanistic inference. (5) Single time-point sampling; toxins are variable day to day. (6) Creatinine normalization assumes stable excretion. (7) No validation cohort. (8) Behavioral severity and GI symptoms not fully recorded. (9) Lacks mechanistic/interventional validation. (10) Reproducibility of MIGD not assessed.

## 4. Materials and Methods

### 4.1. Participants

The study cohort included children diagnosed with autism spectrum disorder (ASD) (*N* = 97) and neurotypical controls (*n* = 71) aged between 2 and 17 years. The inclusion of a broad pediatric age range reflects the natural diagnostic age span for ASD and captures the spectrum of metabolic maturation. Children in the study group were diagnosed with ASD by an expert pediatrician or a neuropsychiatrist in collaboration with a psychologist. A multidisciplinary team consisting of pediatricians, psychiatrists, and psychologists diagnosed ASD with a clinical assessment and a psychological assessment. The ASD criteria summarized by the DSM-5 were used [46]. The cohort in this study is independent from the population described in [8], and no participant overlap occurred.

The study protocol was approved by the National Medical Ethics Committee (0120-201/2016-2 KME 78/03/16).

Informed consent was obtained from legal guardians, and the study was approved by the relevant ethics committee. Sample Size Considerations: Study used a convenience sample (97 ASD, 71 controls). No a priori calculation; post hoc analysis indicates adequate power for major comparison but low power for phenotype subgroups. 50+ subjects per group recommended for future studies.

### 4.2. Stool Assessment

Stool consistency was classified according to the Bristol Stool Chart (BSC), with types 1–2 designated as hard stools, 3–5 as normal, and 6–7 as loose stools. Subgroups were created accordingly within the ASD and control populations.

Table 4 details the distribution of sex and age among ASD and control children by BSC category. A male predominance is noted in all subgroups, especially in those with normal stools (BSC 3–5). Median ages are similar across groups, though slightly higher in those with looser stools. This stratification helps interpret metabolite levels by accounting for age-related renal function and metabolic output, and sex-linked differences in gut microbiota and toxin excretion.

### 4.3. Urine Collection and Normalization

First-morning urine samples were collected using standard procedures. Samples were centrifuged and stored at −80 °C until analysis.

### 4.4. Methods

#### 4.4.1. Toxin Quantification

Urinary concentrations of p-cresyl sulfate (PCS), trimethylamine N-oxide (TMAO), indoxyl sulfate (IS), symmetric dimethylarginine (SDMA), and asymmetric dimethylarginine (ADMA) were quantified using ultra-performance liquid chromatography with tandem mass spectrometry (UPLC-MS/MS), according to validated protocols. The instrumentation used for quantification was the Agilent 1260 Infinity LC system (Agilent Technologies, Santa Clara, CA, USA) interfaced with a QTRAP 4000 mass spectrometer (AB Sciex, Framingham, MA, USA). Each compound was quantified by monitoring two transitions (MRM mode). Reference compounds and stable isotope-labeled internal standards were sourced from Toronto Research Chemicals (TRC, Vaughan, ON, Canada). Analytical sensitivity, reproducibility, and recovery rates met standard criteria for clinical metabolomics.

#### 4.4.2. Quantification of TMAO, ADMA, and SDMA

The analysis was conducted in MRM mode, applying the following transitions with corresponding DP and CE settings: ADMA (m/z 203 → 46; DP: 61 V, CE: 41 V), SDMA (m/z 203 → 172; DP: 61 V, CE: 19 V), ADMA-d6 (m/z 209 → 77; DP: 66 V, CE: 45 V), TMAO (m/z 76 → 42; DP: 66 V, CE: 53 V), and TMAO-d9 (m/z 85 → 46; DP: 61 V, CE: 59 V). Separation was carried out on a SeQuant^®^ ZIC^®^-HILIC column (50 × 2.1 mm, 5 µm; MZ-Analysentechnik GmbH, Mainz, Germany) maintained at 25 °C, with a mobile phase flow rate of 0.5 mL/min. The mobile phases consisted of 20 mM ammonium acetate (solvent A) and acetonitrile with 0.2% formic acid (solvent B). The gradient program was set as follows: 95% B for 0–1 min, decreasing to 50% B at 7–8 min. The injected volume was 5 µL. For sample preparation, 0.1 mL of urine was mixed with 0.1 mL internal standard solution (6 µg/mL ADMA-d6 and TMAO-d9) and 0.6 mL acetonitrile, vortexed for 3 min, centrifuged (10,000× *g*, 5 min), and the supernatant was injected into the LC system.

#### 4.4.3. Quantification of PCS and IS

MRM transitions used were PCS (m/z 186.9 → 106.9), PCS-d7 (m/z 194.0 → 114.0), IS (m/z 211.9 → 79.8), and IS-d4 (m/z 216.0 → 79.9). Instrument settings included the following voltages: for IS and IS-d4—DP/CE/EP/CXP: −60/−38/−10/−5 V and −65/−46/−10/−1 V, respectively; for PCS and PCS-d7—−65/−28/−10/−7 V and −60/−30/−10/−7 V. Chromatographic separation was achieved on a Kinetex C18 column (100 × 4.6 mm, 2.6 µm; Phenomenex, Torrance, CA, USA) maintained at 40 °C with a 0.5 mL/min flow rate. The mobile phase system consisted of 0.1% formic acid in water (A) and in methanol (B). The gradient profile was 10% B from 0 to 0.5 min, ramping to 95% B from 4.5 to 8.5 min. A 10 µL aliquot was injected. For sample processing, 0.01 mL of urine was mixed with 0.05 mL of internal standards (2 µg/mL PCS-d7 and IS-d4) and 0.6 mL of methanol, vortexed for 3 min, centrifuged at 10,000 g for 5 min, diluted with water (6 times), and analyzed by LC-MS/MS. LC-MS/MS methods were validated in accordance with the European Medicines Agency (EMA) guidelines for bioanalytical method validation. Calibration curves were prepared in the following concentration ranges: 1–100 µg/mg for ADMA (r = 0.997), 5–200 µg/mg for SDMA (r = 0.997), 10–200 µg/mg for TMAO (r = 0.992), 0.2–500 µg/mg for PCS (r = 0.997), and 0.5–167 µg/mg for IS (r = 0.999), using a 1/x weighting factor.

The absolute matrix effect was 97%, 95%, 92%, 95%, and 107% for PCS, IS, ADMA, TMAO, and SDMA, respectively, indicating no significant matrix interference. The variation in the relative matrix effect was below 15% for each analyte, meeting the acceptance criteria defined in the EMA guidelines [47].

Creatinine levels were measured enzymatically, and metabolite concentrations were normalized to creatinine (µmol/mmol creatinine) to adjust for urinary dilution.

#### 4.4.4. Data Processing and Interpretation

To enhance interpretability and functional stratification, we classified participants using a novel composite biomarker: the Metabolic Index of Gut Dysfunction (MIGD). We also refer to this biomarker as the Osredkar-Godnov Index (OGI). MIGD/OGI was calculated as
MIGD = (PCS/TMAO × 100) ÷ (IS/ADMA)(1)

This formula integrates phenolic fermentation (PCS/TMAO) and indolic detoxification (IS/ADMA) into a unified scale. The 100 multiplier was applied to optimize readability and visualization. Higher MIGD values indicate more pronounced metabolic imbalance, especially with high phenolic and low indolic pathway efficiency.

For MIGD, we used previously established biological logic and observed distributional patterns in our cohort to propose the interpretive ranges (see Table 5). Ranges are provisional, based on quartiles and biological expectation; not yet validated externally.

In practice:

MIGD/OGI is usually calculated using group-level ratios or sometimes individual-level averages, or adjusted for visualization (e.g., log-transformed).

PCS, TMAO, IS, and ADMA may not be normally distributed and may be log-transformed, trimmed, or rescaled differently across indices.

In group-wise summaries, small denominator values (e.g., TMAO or ADMA near zero) can make MIGD unstable and inflate non-linearities.

MIGD was calculated both group-level (using subgroup medians for Tables) and individual-level (for statistical analysis, e.g., in Table 2). Individual-level calculations omitted subjects with near-zero denominators.

### 4.5. Statistics

R version 4.3.1, in conjunction with RStudio version 2023.12.0, was used for statistical analysis, using the tidyverse suite [48] for visualization and the arsenal package [49] to compare groups.

The Shapiro–Wilk test was used to evaluate the distribution of data. Results showed that all datasets had non-normal distributions, so the non-parametric Kruskal–Wallis test was employed for comparisons. The Benjamini–Hochberg procedure was used to control the false discovery rate, with an alpha significance level of 0.05.

MIGD Formula Application: The Metabolic Index of Gut Dysfunction (MIGD) was calculated using the formula (PCS/TMAO × 100) ÷ (IS/ADMA). Medians of the respective toxins were used for this composite index. Additional ratios (PCS/TMAO and IS/ADMA) were computed to support the interpretation of microbial fermentation dynamics.

All statistical analyses and visualizations were performed using GraphPad Prism version 10.2.0 and R statistical software 4.5.

## 5. Conclusions

This study presents an exploratory framework for characterizing gut-derived metabolic profiles in children with ASD using a novel composite biomarker derived from urinary uremic toxins. While group-wise statistical differences were not consistently significant, clear **directional trends** emerged when stratifying by stool phenotype. The Metabolic Index of Gut Dysfunction (MIGD), also called the Osredkar-Godnov Index (OGI) here, provides an integrative measure of microbial–host metabolic imbalance by combining phenolic and indolic pathway dynamics. In this study, MIGD/OGI revealed distinct though non-significant directional trends across ASD subgroups stratified by stool consistency, with higher values in children with hard stools and lower values in those with loose stools. These patterns support the hypothesis that gut microbial fermentation in ASD varies with bowel habits and transit time. While exploratory, these trends indicate MIGD/OGI may stratify ASD subtypes by gut metabolic profile. Further validation in larger, longitudinal cohorts, with microbiome and dietary assessment, is essential before clinical use.

## Figures and Tables

**Figure 1 ijms-26-10475-f001:**
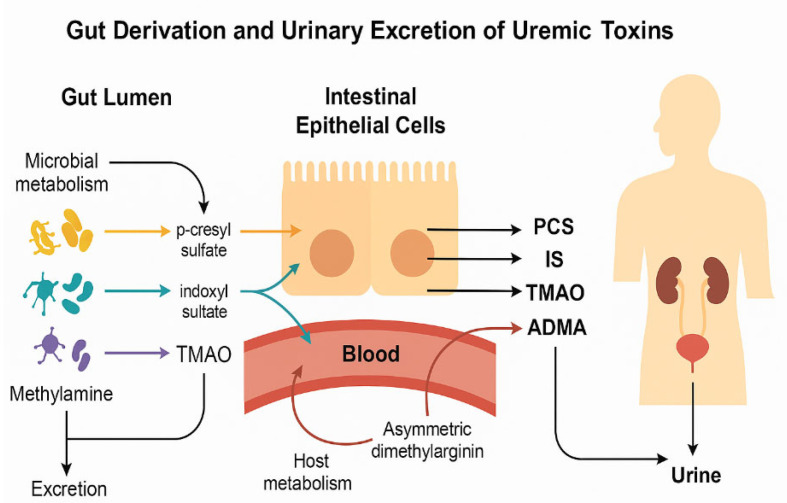
Microbial metabolism of dietary compounds and formation of uremic toxins in ASD. Conceptual illustration of the origin and urinary excretion of uremic toxins relevant to ASD. Microbial metabolism in the gut lumen produces: *Clostridium difficile* PCS from tyrosine, *E. coli*, *Bacteroides* IS from tryptophan, and TMAO from hepatic FMO3 oxidation of TMA from choline/carnitine by gut bacteria. These toxins cross the intestinal epithelium into the bloodstream and are subsequently filtered into urine. ADMA, generated via host metabolism, reflects nitric oxide inhibition and endothelial stress. All four toxins were quantified in urine samples and used to derive functional indices of gut metabolic imbalance in ASD children.

**Table 1 ijms-26-10475-t001:** Urinary uremic toxin concentrations (µmol/mmol creatinine) across ASD and control groups stratified by Bristol Stool Chart (BSC) type.

	Control (*N* = 71)	ASD (All) (*N* = 97)	*p* Value	1 (BSC 1,2) (*N* = 28)	1 (BSC 6,7) (*N* = 8)	1 (BSC 3,4,5 (*N* = 61)	*p* Value			p TOT
	0	1	0:1	2	3	4	0:2	0:3	0:4	
ADMA			0.48				0.68	0.28	0.25	0.356
Median (Q1, Q3)	12.70 (9.73, 18.74)	14.76 (10.50, 17.03)		14.79 (10.69, 16.95)	8.02 (7.57, 15.15)	15.09 (11.32, 19.54)				
SDMA			0.62				0.76	0.48	0.88	0.573
Median (Q1, Q3)	31.32 (23.01, 38.09)	29.06 (20.10, 39.77)		29.09 (20.17, 36.91)	19.73 (14.64, 50.79)	29.08 (23.16, 44.29)				
TMAO			0.51				0.11	0.29	0.76	0.115
Median (Q1, Q3)	3.09 (1.48, 5.19)	2.64 (1.64, 5.01)		2.38 (1.69, 3.52)	2.60 (0.04, 3.81)	3.19 (1.67, 5.53)				
IS			0.25				0.43	0.91	0.26	0.869
Median (Q1, Q3)	63.80 (39.77, 103.18)	56.77 (28.22, 89.25)		52.26 (41.88, 79.10)	89.25 (23.98, 108.77)	61.52 (27.95, 89.20)				
PCS			0.53				0.20	0.92	0.87	0.705
Median (Q1, Q3)	37.74 (20.02, 80.02)	51.84 (17.07, 86.81)		46.65 (29.01, 86.73)	31.85 (9.09, 140.45)	52.65 (15.80, 84.76)				

ADMA = asymmetric dimethylarginine; SDMA = symmetric dimethylarginine; TMAO = trimethylamine N-oxide; IS = indoxyl sulfate; PCS = p-cresyl sulfate. Concentrations are expressed as µmol per mmol creatinine. ASD subgroups stratified by Bristol Stool Chart: BSC 1–2 (constipation), BSC 3–5 (normal), BSC 6–7 (loose). Kruskal–Wallis test with Benjamini–Hochberg correction.

**Table 2 ijms-26-10475-t002:** Composite indices of gut metabolic dysfunction (MIGD) in controls and ASD subgroups stratified by stool consistency.

	Control (*N* = 71)	ASD (All) (*N* = 97)	*p* Value	1 (BSC 1,2) (*N* = 28)	1 (BSC 6,7) (*N* = 8)	1 (BSC 3,4,5) (*N* = 61)	*p* Value			p TOT
	0	ASD	0:ASD	1	2	3	0:1	0:2	0:3	
MIGD			0.20				0.49	0.10	0.08	0.088
Median (Q1, Q3)	316.05 (144.24, 475.01)	342.33 (163.65, 752.00)		342.33 (172.99, 854.95)	140.02 (104.96, 271.75)	374.49 (194.61, 732.45)				

**Table 3 ijms-26-10475-t003:** MIGD and component toxin ratios calculated from median values.

Group	PCS/TMAO	IS/ADMA	MIGD
Controls (BSC 3–5)	12.4	5.2	238.5
ASD (BSC 3–5)	16.5	4.1	402.4
ASD (BSC 6–7)	12.3	11.1	110.8
ASD (BSC 1–2)	19.6	3.5	560.0

**Table 4 ijms-26-10475-t004:** Demographic distribution of participants by sex and age within stool consistency groups.

	Control (*N* = 71)	1 (All) (*N* = 97)	1 (BSC 1,2) (*N* = 28)	1 (BSC 6,7) (*N* = 8)	1 (BSC 3,4,5) (*N* = 61)
SEX					
Boys	37 (52.1%)	76 (78.4%)	17 (60.7%)	5 (62.5%)	54 (88.5%)
Girls	34 (47.9%)	21 (21.6%)	11 (39.3%)	3 (37.5%)	7 (11.5%)
AGE (years)					
Mean (SD)	8.93 (3.82)	9.44 (3.77)	9.76 (4.21)	9.88 (3.15)	9.24 (3.68)
Median (Q1, Q3)	8.60 (6.20, 11.25)	8.70 (6.20, 12.40)	9.05 (6.18, 13.03)	10.40 (8.05, 12.55)	8.70 (6.20, 11.30)
Min–Max	2.40–16.70	2.50–17.00	3.50–17.00	4.60–13.20	2.50–16.70

Control group—0; ASD group—1.

**Table 5 ijms-26-10475-t005:** Interpretation ranges for the composite biomarker MIGD/OGI.

MIGD Value Range	Interpretation
<50	Low metabolic disruption or compensatory indole pathway activity
50–150	Mild to moderate metabolic imbalance
150–300	Marked microbial–host metabolic disturbance
>300	High dysfunction—skewed fermentation and impaired detoxification
>500	Severe imbalance—indicates high systemic fermentation burden

## Data Availability

The data that support the findings of this study are available from the study’s principal investigator—O.J.—upon reasonable request.

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
