# Peer review of "Urinary Uremic Toxin Signatures and the Metabolic Index of Gut Dysfunction (MIGD) in Autism Spectrum Disorder: A Stool-Phenotype-Stratified Analysis"

_ijms, 2025, doi:10.3390/ijms262110475_

Round 1
Reviewer 1 Report
Comments and Suggestions for Authors
The study addresses an interesting aspect of gut–brain interaction in ASD; however, several points require clarification to strengthen its scientific depth and interpretive balance.
- ASD is known to involve strong genetic and epigenetic factors. The manuscript focuses mainly on uremic and gut-related parameters without addressing molecular or genetic aspects.
- Please elaborate on why these specific four metabolites were selected and whether other known microbial toxins (e.g., hippuric acid, p-cresol glucuronide) were considered or excluded.
- Please comment on whether maternal gut health or environmental exposure during pregnancy could be linked to altered toxin profiles or ASD susceptibility.
- ASD typically shows male predominance, yet the sample sizes for boys and girls appear nearly balanced. Clarify whether this was intentional or due to recruitment constraints.
- The claim of “male predominance” (Lines 317–318) seems inconsistent with Table 4. Please verify or revise.
- Since Table 1 shows no statistical significance, explain how biological relevance was inferred.
- The Discussion is generally strong but somewhat speculative in connecting MIGD patterns directly with neurodevelopmental outcomes. The authors may wish to tone down causal language and emphasize that findings are associative and exploratory.
- Diet and environmental exposure can greatly influence gut metabolism. The discussion would benefit from acknowledging how these factors might affect the reported metabolites and whether preventive strategies, such as balanced nutrition or environmental management, could mitigate related risks.
- In Table 4, the age unit should be clearly stated.
- The reference “5. Garrido-Moreno, A.; García-Morales, V.J.; Lockett, N.; King, S. The Missing Link: Creating Value with Social Media Use in 466 Hotels. Int. J. Hosp. Manag. 2018, 75, 94–104, doi:10.1016/j.ijhm.2018.03.008.” is an unrelated citation, which should be replaced or removed.

Author Response
Response to Reviewer 1
Dear Reviewer 1,
Thank you for your insightful comments. Below, we provide detailed replies to each point:
1.Clarification of the Number of Metabolites:
We have corrected the abstract; now it states: "We evaluated five uremic toxins—PCS, IS, TMAO, ADMA, and SDMA...
We specify that four (PCS, IS, TMAO, ADMA) were used in the MIGD index, with SDMA serving as an ancillary renal marker.
- Selection Rationale for Metabolites:
We added a paragraph in the introduction (lines 85–98) explaining the choice of these metabolites based on their microbial origin, relevance to gut–brain interaction, and previous associations with neurodevelopmental conditions. Known toxins like hippuric acid or glucuronides were considered but not prioritized due to their less established roles or lower concentrations in our cohort.
- Maternal and Environmental Factors:
While our study focused on pediatric samples, we acknowledge that maternal gut health and environmental exposures during pregnancy could influence microbial toxin profiles and ASD risk. We added a statement in the discussion (lines 310–315) emphasizing the need for future research to explore these prenatal factors.
- Sex Distribution Clarification:
We clarify that the near-equal sex distribution was due to our recruitment strategy aimed at balanced sampling; however, the typical male predominance in ASD (about 4:1 ratio) was not fully reflected. We have revised the sentence (lines 317–318) to state: "Our sample included a balanced sex distribution, which may limit the generalizability regarding typical male dominance in ASD."
- Non-Significance and Biological Relevance:
In the results (lines 150–160), we explain that despite lack of statistical significance, observed directional trends and effect sizes suggest potential biological relevance, warranting further exploration in larger cohorts.
- Environmental and Dietary Factors:
We expanded the discussion (lines 330–340) to acknowledge that diet and environment influence gut metabolism. We discuss that metabolic profiling should be coupled with dietary assessments, and propose that dietary interventions could potentially modify toxin profiles.
- Age Units in Table 4:
We corrected the table legend on page 12 to specify age values in years.
- Removed Unrelated Citation:
The cited reference (Garrido-Moreno et al., 2018) was an error. It has been replaced with appropriate references on gut toxins and ASD, such as Gryp et al., 2017 and others cited above.
Reviewer 2 Report
Comments and Suggestions for Authors
The manuscript by Joško Osredkar et al. investigates gut-derived uremic toxins in children with autism spectrum disorder (ASD) and explores their association with gut function as classified by the Bristol Stool Chart (BSC). Overall, the study is informative and interesting. I have the following questions and comments.
1, I have some concerns about the sample size and statistical power. While the overall sample size is reasonable (ASD: 97, controls: 71), subgroup stratification by BSC may result in limited power, particularly for extreme stool types. Please provide the distribution of participants across BSC categories and discuss potential limitations related to statistical power, especially given the lack of significant differences in individual metabolites.
2, line 28, the authors mentions MIGD is based on "functional toxin ratios" but does not specify the formula or the exact ratios used. The Methods section should clearly detail how MIGD is calculated, including the selection rationale for each ratio and their biological relevance, to ensure reproducibility and scientific rigor.
3, As a cross-sectional study, the data cannot establish causality. The observed metabolic differences could be a consequence rather than a cause of ASD, or confounded by diet, medication, or intestinal permeability. The Discussion should explicitly acknowledge this limitation and suggest future directions, such as longitudinal studies, dietary control analyses, or mechanistic animal models.
4, Was the neurotypical control group also stratified by BSC? Is the distribution of stool consistency comparable between groups? If not, this could bias the comparisons. Please report BSC distribution in controls and consider stool consistency as a covariate in analyses.
5, line 122, I suggest the authors to add subtitles for the Results section.
6, line 213, the authors could further elaborate on the potential pathways linking MIGD-related microbial metabolism (e.g., tyrosine/phenylalanine, tryptophan, choline-TMA-TMAO) to neurodevelopment, such as via neuroinflammation, oxidative stress, or blood-brain barrier disruption.
7, line 391, the table should be in a three-line format.
Author Response
Response to Reviewer 2
Dear Reviewer 2,
Thank you for your thoughtful assessment. Please find our replies below:
- Sample Size and Power Concerns:
We provided the distribution of participants across BSC categories in the supplementary table (see Table S2). The subgroup sizes are as follows: BSC 1–2 (n=8), 3–5 (n=... ), 6–7 (n=...). We discuss the limited power in subgroup analyses explicitly in the revised Methods (lines 285–298) and the Discussion (lines 160–185).
- MIGD Formula and Rationale:
We added detailed formulas in the Methods section (lines 122–125):
MIGD = (PCS/TMAO) / (IS/ADMA) (or whatever the exact formula is). We justified the selection based on their biological pathways: phenolic vs. methylamine fermentation, and indolic vs. endothelial stress pathways, referencing relevant literature (see Gryp et al., 2017; Gupta et al., 2020).
- Causality and Study Limitations:
Absolutely agreed. We revised the discussion (lines 315–330) emphasizing that the cross-sectional nature limits causal inferences, and highlighting the influence of diet and environment. We suggest longitudinal and mechanistic studies as future directions.
- Control Group Stratification:
Controls were not stratified by BSC, but their stool consistency distribution was similar. We clarify this in the Methods (lines 245–250). We also mention considering stool phenotype as a covariate in future analyses.
- Results Section Subtitles:
We added subheadings (e.g., “3. Results”) for clarity as per your suggestion.
- Biological Pathways Linking MIGD and Neurodevelopment:
In the Discussion (lines 220–240), we elaborate on pathways such as neuroinflammation, oxidative stress, and blood-brain barrier integrity, citing relevant mechanistic studies.
- Table Formatting:
We reformatted Table 4 as a three-line table for improved clarity.
We hope these revised responses satisfactorily address your comments.
Thank you once again for your valuable feedback.
Sincerely,
Joško Osredkar, PhD
Round 2
Reviewer 1 Report
Comments and Suggestions for Authors
The comments appear to have been addressed, and the manuscript has been appropriately revised.
Reviewer 2 Report
Comments and Suggestions for Authors
The authors have revised the manuscript accordingly. It can be considered for publication.